# Effects of Microwave Sintering Temperature and Holding Time on Mechanical Properties and Microstructure of Si_3_N_4_/n-SiC ceramics

**DOI:** 10.3390/ma12233837

**Published:** 2019-11-21

**Authors:** Li Qiao, Zhenhua Wang, Taiyi Lu, Juntang Yuan

**Affiliations:** 1School of Mechanical Engineering, Nanjing University of Science & Technology, Nanjing 210094, Chinamc106@njust.edu.cn (J.Y.); 2Collaborative Innovation Center of High-End Equipment Manufacturing Technology (Nanjing University of Science & Technology), Ministry of Industry and Information Technology, Nanjing 210094, China

**Keywords:** Si_3_N_4_ ceramics, n-SiC, microwave sintering, mechanical properties, microstructure

## Abstract

The n-SiC (nanometer SiC) is added to be the additive in order to improve the mechanical performance of Si_3_N_4_ ceramics. A microwave sintered the ceramics at different temperature and holding times. The results shows that the Si_3_N_4_/n-SiC ceramics (85 wt% Si_3_N_4_ + 5 wt% n-SiC + 5 wt% Al_2_O_3_ + 5 wt% Y_2_O_3_) have the best mechanical properties at 1600 °C, which is beneficial to the densification and β-Si_3_N_4_ phase formation for 10 min: the density, hardness, and fracture toughness were 97.1%, 14.44 GPa, and 7.77 MPa·m^1/2^, which increased by 2.8%, 7.0%, and 13.1%, respectively, when compared with the ceramics (90 wt% Si_3_N_4_ + 5 wt% Al_2_O_3_ + 5 wt% Y_2_O_3_).

## 1. Introduction

Si_3_N_4_ ceramic have been widely recognized as the ideal tool materials for cutting the high temperature cast iron and nickel-based alloys due to its outstanding thermal properties, wear resistance, and high temperature material properties [1,2,3,4]. Currently, high-performance densified Si_3_N_4_ ceramics are usually sintered by hot pressing sintering [5,6], hot isostatic pressure sintering [7], pressure less sintering [8], microwave sintering [3,4], reaction sintering [9,10], and spark plasma sintering [11,12,13]. Among the mentioned methods, microwave sintering is a new technology, which has high-energy utilization rate, safety, and free pollution. Many researches have shown that it can not only promote the densification of sintered materials to short the sintering time, but also refine the crystalline microstructure [14,15,16,17]. Besides, it is a kind of pressure less sintering, which makes it possible to massively produce ceramic tools with complex shapes.

However, the microwave absorption of Si_3_N_4_ at low temperature (less than 800 °C) is poor [18], which affects the microwave sintering process. The fracture toughness of Si_3_N_4_ ceramic is usually between 5.0 MPa·m^1/2^ and 7.0 MPa·m^1/2^ with high hardness [3,13,14,15,16,17], which should be improved to meet the requirements of high-performance cutting tools. When considering that n-SiC has good microwave absorption properties and high hardness at all temperature stages [18], it is added into silicon nitride ceramics. On the one hand, the uniform distributed n-SiC particles can be used as strengthening phase to improve the fracture toughness of the ceramics. On the other hand, the n-SiC particles are equivalent to many small heat sources during the sintering process, which improve the sintering properties and promote the α→β phase transformation of the Si_3_N_4_.

Al_2_O_3_ and Y_2_O_3_ were the most common additives that were used in the conventional sintering of Si_3_N_4_ ceramics according to the existing reports [19,20,21,22,23]. In the field of microwave sintering, Al_2_O_3_ and Y_2_O_3_ were also used as sintering additives, for example, Tiegs et al. [24] successfully sintered silicon nitride ceramic material with the density of 96% by microwave sintering while using 12 wt% Y_2_O_3_ and 4 wt% Al_2_O_3_ as sintering additives. Yoon et al. [14] fabricated Si_3_N_4_ ceramic materials with a density of 99% at a sintering temperature of 1600 °C while using Al_2_O_3_ and Y_2_O_3_ as sintering additives. As a result of that these two kinds of sintering additives can contribute to the formation of the liquid phase, which can promote the densification of Si_3_N_4_ ceramics, we selected these sintering aids in our research. The study was aimed to prepare high-performance Si_3_N_4_/n-SiC ceramics, with Al_2_O_3_ and Y_2_O_3_ as sintering aids, by microwave. The effects of sintering temperature and holding time on the properties of microwave sintered Si_3_N_4_/n-SiC ceramics were analyzed. SEM microstructure analysis, X-ray diffraction (XRD) phase analysis, density, and hardness and fracture toughness. 

## 2. Experimental Procedure 

The starting powders (99.9% purity, Shanghai, China) were used as: α-Si_3_N_4_ (0.7 μm), n-SiC (0.1 μm), Y_2_O_3_ (1 μm), and Al_2_O_3_ (0.5 μm). Firstly, the starting powders that were in isopropyl alcohol were placed on a planetary ball mill (QM-3SP2, Nanjing university instrument factory, Nanjing, China) for 48 h. Subsequently, the 3 wt% polyvinyl alcohol solution was added to the mixed slurry and the ball milling was continued to ball mill for 2 h. After that, the powders were then pressurized under 180 MPa for 2 min and finally turned into a green body that has a certain shape and density, and it was sintered in a multimode high temperature vacuum microwave oven filled with N_2_. Figure 1 shows the heat preservation device for sample. Table 1 lists the compositions, sintering temperature, and holding time of Si_3_N_4_ materials. T 90 wt% Si_3_N_4_ + 5 wt% Al_2_O_3_ + 5 wt% Y_2_O_3_ is taken as a comparison to illustrate the effectiveness of n-SiC on the properties of Si_3_N_4_ ceramics.

The density of the sintered specimens was measured by the Archimedes drainage method with distilled water. According to the method in [25], the grain size analysis is carried on. After grinding and polishing by a diamond quadrilateral indenter (HV-50, Insize, Shanghai, China) under 196.1 N load the Vickers hardness (Hv) was tested. Fracture toughness was determined while using the indentation technique in [26]. The phase composition of the sample before and after sintering was analyzed via X-ray diffractometric (D8 Advance, Bruker, Karlsruhe, Germany). The relative amounts of the α-Si_3_N_4_ and β-Si_3_N_4_ phases that were presented in the Si_3_N_4_ specimen were determined according to the peak height [27]. The microstructures of the samples were observed by Scanning electron microscopy (SEM, FEI Quanta 250F, FEI, Hillsboro, OR, USA).

## 3. Results and Discussion

### 3.1. Effect of Sintering Temperature 

Figure 2 shows the he relative density increases first and then decreases with the increase of sintering temperature, the minimum relative density reached the minimum value 95.5%, and the maximum value 97.1% at 1400 °C and 1600 °C, respectively. As shown in Figure 3a, there are still many porosities in the sintered Si_3_N_4_/n-SiC ceramics materials at 1400 °C. This is mainly because enough liquid phases inside the sintered ceramics material cannot be formed to complete sintering at a low temperature. Figure 3b–d indicate that the porosity of the sintered Si_3_N_4_/n-SiC ceramics material was obviously reduced with as the sintering temperature increases and the material structure is more compact than that at 1400 °C. The results show that the sintering additives (5 wt% Al_2_O_3_ and 5 wt% Y_2_O_3_) were able to form a sufficient liquid phase with the increase of temperature during the sintering process. The sufficient liquid phase further promotes the particles motion (Si_3_N_4_ and n-SiC) and the shrinkage of the green compact, thus the densification of sintered ceramic materials was improved. However, when the sintering temperature reaches to 1700 °C, the relative density of sintered ceramic materials slightly decrease. It can be explained that, with the increase of sintering temperature, the β-Si_3_N_4_ phase gradually increases and grows, and the interlocking structure is formed, which causes the increase of the porosity and the decrease of the relative density. Another possible explanation might be the errors introduced in the process of temperature measurement. High temperature (over 1800 °C) will cause the decomposition of silicon nitride, which leads to the decrease of relative density [28]. In addition, due to the good microwave absorbing property of SiC, the temperature of the n-SiC particles is higher than the Si_3_N_4_ particles’ during microwave sintering; therefore, a local high temperature region is formed around the n-SiC particles. The local high temperature can promote the densification of sintered material and the α→β phase transformation of Si_3_N_4_. However, if the local temperature is above 1800 °C, the Si_3_N_4_will decompose.

Figure 4 shows the SEM micrographs of the sintered Si_3_N_4_/n-SiC ceramics at different temperatures. Figure 5 shows the XRD results of the sintered Si_3_N_4_/n-SiC materials. As it can be seen in Figure 4a, there is a small amount of long columnar structure and lots of powder or irregular shape, which is formed by grains of sintered material, when the temperature is 1400 °C. Figure 5a shows that the sintered samples mainly consist of α-Si_3_N_4_, β-Si_3_N_4_, SiC, and Si_5_AlON_7_ at 1400 °C. The ratios of α-Si_3_N_4_ and β-Si_3_N_4_ in the Si_3_N_4_ ceramics are 52% and 48%, respectively. This further indicates that there are not enough liquid phases, which are used to promote the sintering densification and the transformation of α-Si_3_N_4_ to β-Si_3_N_4_, at a relatively low sintering temperature. As shown in Figure 4b, most of the α-Si_3_N_4_ is transformed into a long columnar β-Si_3_N_4_ at 1500 °C, and there is only a small amount of α-Si_3_N_4_ in the sintered Si_3_N_4_/n-SiC ceramics, as shown in Figure 5. However, the crystal size of β-Si_3_N_4_ is different, there are some silicon nitride grains that grow more completely, and presenting larger grains size, and there are also many small columnar grains. As can be seen from Figure 4c, the grains of β-Si_3_N_4_ were further grown and formed a large number of big columnar grains with an average grain length of 3.25 μm at 1600 °C. As shown in Figure 4d, the surface of some β-Si_3_N_4_ grains is not smooth, and there is especially the tendency that some β-Si_3_N_4_ grains are going to turn to powder, when the temperature is 1700 °C. It is further indicated that Si_3_N_4_ has the possibility of decomposition at this temperature, which reduces the mechanical properties and densification of the sintered Si_3_N_4_/n-SiC ceramics. According to the XRD diffraction pattern (Figure 5), the α-Si_3_N_4_ phase is completely transformed into a β-Si_3_N_4_ phase in sintered Si_3_N_4_/n-SiC ceramics when the sintering temperature is above 1600 °C. 

As shown in Figure 6, the hardness and fracture toughness increase first and then decrease as the sintering temperature increased. The change trend of the Vickers hardness and fracture toughness is the same as relative density. It indicates that the high densification can significantly improve the mechanical performances of the sintered Si_3_N_4_/n-SiC ceramics. When the temperature is 1600 °C, the Vickers hardness and fracture toughness reached a maximum value of 14.44 GPa and 7.77 MPa·m^1/2^, respectively. As can be seen from Figure 3c and Figure 4c, the sintered Si_3_N_4_/n-SiC ceramics were densified and formed many interlocking elongated columnar β-Si_3_N_4_ grains. The main fracture mechanisms of the sintered Si_3_N_4_/n-SiC ceramics are intergranular fracture, transgranular fracture, and grain pulling out. The interlocking microstructure that formed by large elongated columnar β-Si_3_N_4_ grains can greatly enhance the mechanical properties, especially the fracture toughness of silicon nitride materials. Therefore, when the temperature exceeds 1500 °C, a great quantity of elongated columnar β-Si_3_N_4_ grains is formed, which significantly improves the fracture toughness (more than 7.0 MPa·m^1/2^) of the material.

Table 2 shows the comparison of properties of Si_3_N_4_ with n-SiC content of 0% and 5% at 1600 °C for 10 min. When n-SiC is not added into the sintered silicon nitride materials, the density, Vickers hardness, and fracture toughness of the Si_3_N_4_ ceramics are 94.5%, 13.5 GPa, and 6.87 MPa·m^1/2^, which increased by 2.6%, 7.0%, and 13.1%, respectively. Figure 7 is SEM micrographs of the fracture surface morphology of the Si_3_N_4_ material with 0% n-SiC content. As shown in Figure 7, when the n-SiC content is 0%, there are great quantities of small porosities in the sintered Si_3_N_4_ ceramics. Additionally, the distribution and size of the β-Si_3_N_4_ grains are uneven. The results showed that the filling of SiC nanopowders evidently improved the microwave sinter ability of Si_3_N_4_ ceramics.

### 3.2. Influence of Holding Time on the Properties of Si_3_N_4_/n-SiC Ceramics 

Figure 8 shows the influence of holding time on the density of the sintered Si_3_N_4_/n-SiC ceramic at 1600 °C. As shown in Figure 8, the relative density increases first and then decreases as the holding time increased. The density reaches the maximum value of 97.1% when the holding time is 10 min. Figure 3c shows that the microstructure of the sintered Si_3_N_4_/n-SiC ceramic is more compact at this condition. Figure 9 presents SEM micrographs of fracture surface morphology of the Si_3_N_4_/n-SiC ceramic at 1600 °C under different holding time. As shown in Figure 9a, the microstructure of the sintered Si_3_N_4_/n-SiC ceramic is loose when the holding time is 0 min. It can be seen that the sintered Si_3_N_4_/n-SiC ceramic is not completely sintered under this condition. As shown in Figure 8b,c, there are many evenly distributed porosities in the sintered Si_3_N_4_/n-SiC ceramic when the holding time is 20 min and 30 min, respectively. The main reason is that the elongated columnar β-Si_3_N_4_ grains grow with the increase of holding time, which results in a large number of porosities among the big β-Si_3_N_4_ grains.

Figure 10 presents SEM micrographs of the sintered Si_3_N_4_/n-SiC ceramics at different holding time. Figure 11 is the XRD results of Si_3_N_4_/n-SiC ceramics at different holding time. As shown in Figure 10a, when the holding time is 0 min, most of the grains in the sintered Si_3_N_4_/n-SiC ceramics are spherical and powdery, and only a small amount of grains are columnar grains. The average grain size is 0.92 μm, which is nearly the same as that of the starting α-Si_3_N_4_ powder (0.7 μm). Figure 11a shows that the Si_3_N_4_/n-SiC ceramics material is not completely sintered due to the short holding time (0 min). As shown in Figure 4c and Figure 11b, the α-Si_3_N_4_ phase is almost completely transformed into the elongated columnar β-Si_3_N_4_ phase in the sintered Si_3_N_4_/n-SiC ceramics when the holding time is 10 min. As shown in Figure 10b,c, the distribution of β-Si_3_N_4_ crystal size is uneven when the holding time is more than 10 min, the reasonable explanations is that someβ-Si_3_N_4_ grains grow too large to occupy the growing space for other grains. Figure 10c shows that there are some larger holes or pits on the etched surface of the sintered Si_3_N_4_/n-SiC ceramics. It is a possible explanation that the liquid phase is connected to one piece for the long holding time, which causes some Si_3_N_4_ grains to fall off with the dissolution of the liquid phase during the molten NaOH corrosion process to form holes or pits.

Figure 12 shows the influence of different holding time on the Vickers hardness and fracture toughness of Si_3_N_4_/n-SiC ceramics at a constant temperature of 1600 °C. As can be seen from it, the hardness and fracture toughness are both gradually rising first and then falling over time, and it is clear that they both reach their maximum value, 14.44 GPa and 7.77 MPa·m^1/2^, respectively, after the ceramics having been sintered for 10 min. It turns out that the very first ten minutes of the thermal insulation stage plays a key role in improving the microstructure and mechanical performances of the microwave sintered Si_3_N_4_/n-SiC ceramics, while a longer holding time, which is more than 10 min, will result in abnormal grain growth, uneven grain size and pores, and finally lead to a degradation in material properties, according to the experiments.

Table 3 lists the relative density and mechanical performances of other Si_3_N_4_ materials with different sintering aids and sintering methods. It can be seen that the results vary in not only the different sintering methods based sintering processes, but also the processes using the same sintering method. When compared with the listed results in Table 3, the fracture toughness of newly proposed microwave sintered Si_3_N_4_/n-SiC ceramics have obtained an increase of 12.8–27.4%. It is safe to say that preparing high-performance ceramic materials is of great significance with the rapid development of microwave sintering technology.

## 4. Conclusions

In this paper, the microwave sintering process of the Si_3_N_4_/n-SiC ceramics (85 wt% Si_3_N_4_ + 5 wt% n-SiC + 5 wt% Al_2_O_3_ + 5 wt% Y_2_O_3_) was studied, focusing on the effects of a different temperature and holding time on the properties and microstructure. The major conclusions that are derived from this study are as follows:A certain sintering temperature around 1600 °C is necessary for the densification and β-Si_3_N_4_ phase formation of the Si_3_N_4_/n-SiC ceramics. Held at a temperature below 1500 °C, the Si_3_N_4_/n-SiC ceramics cannot be completely sintered. However, a higher temperature over 1700 °C, can cause the decomposition of Si_3_N_4_ or Si_3_N_4_’s reacting with sintering aids (Al_2_O_3_ and Y_2_O_3_), which will lead to the decrease of density, Vickers hardness, and fracture toughness of the Si_3_N_4_/n-SiC.The density, hardness, and fracture toughness will all gradually degrade if the Si_3_N_4_/n-SiC ceramics are sintered with microwave exceeding 10 min. Although the long holding time makes some β-Si_3_N_4_ grains grow larger, it also causes the uneven size of the β-Si_3_N_4_ grain, and pores appear between the big grains at the same time. Accordingly, it can be concluded that the microwave sintered Si_3_N_4_/n-SiC ceramics should not be held for too long. In this paper, the optimum holding time of microwave sintered Si_3_N_4_/n-SiC ceramics is presented to be 10 min at 1600 °C.The microwave sintered Si_3_N_4_/n-SiC ceramics have the best mechanical performances at a sintering temperature of 1600 °C for 10 min, with the density being 97.1%, the Vickers hardness being 14.44 GPa, and the fracture toughness reaching 7.77 MPa·m^1/2^, which increases by 2.8%, 7.0%, and 13.1%, respectively, when compared with the microwave sintered Si_3_N_4_ ceramics (90 wt% Si_3_N_4_ + 5 wt% Al_2_O_3_ + 5 wt% Y_2_O_3_).

## Figures and Tables

**Figure 1 materials-12-03837-f001:**
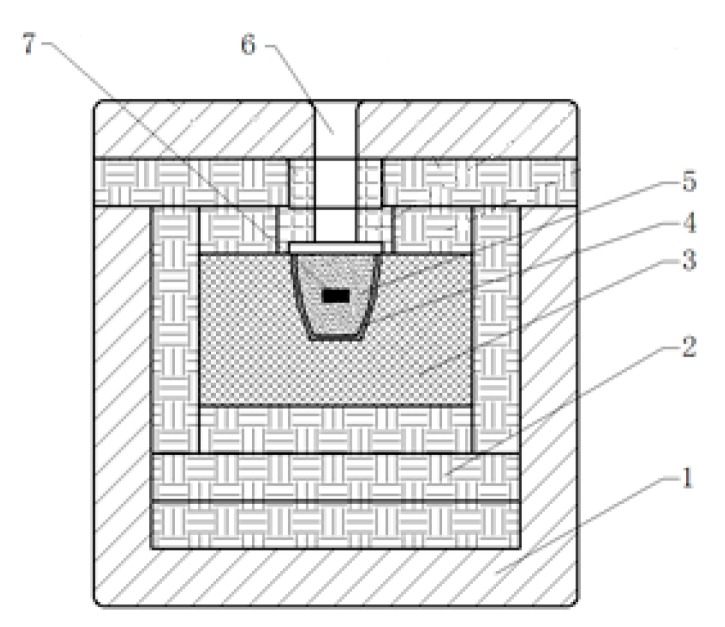
Heat preservation device (1—Mullite box, 2—Aluminium silicate, 3—Polycrystalline mullite fiber cotton, 4—Crucible, 5—Silicon carbide powder, 6—Hole for infrared temperature, and 7—Specimen).

**Figure 2 materials-12-03837-f002:**
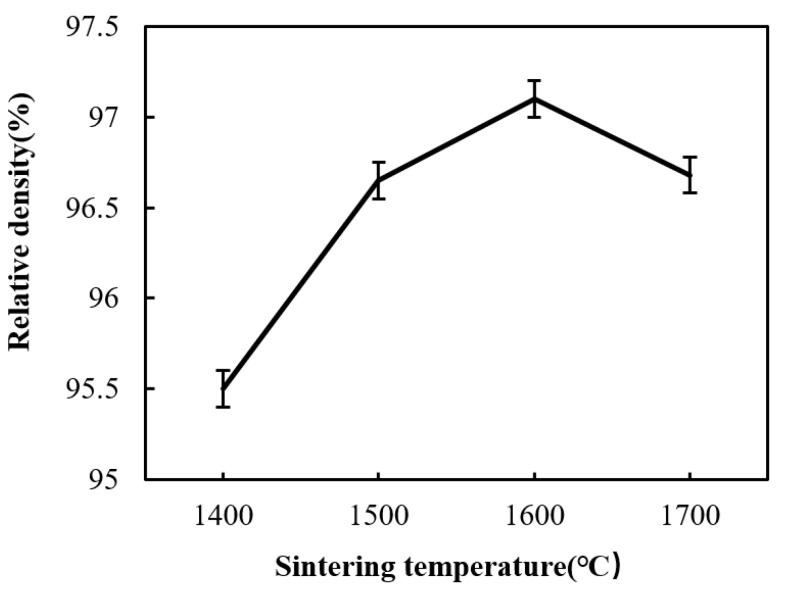
The relationship between the relative density and sintering temperature (holding for 10 min).

**Figure 3 materials-12-03837-f003:**
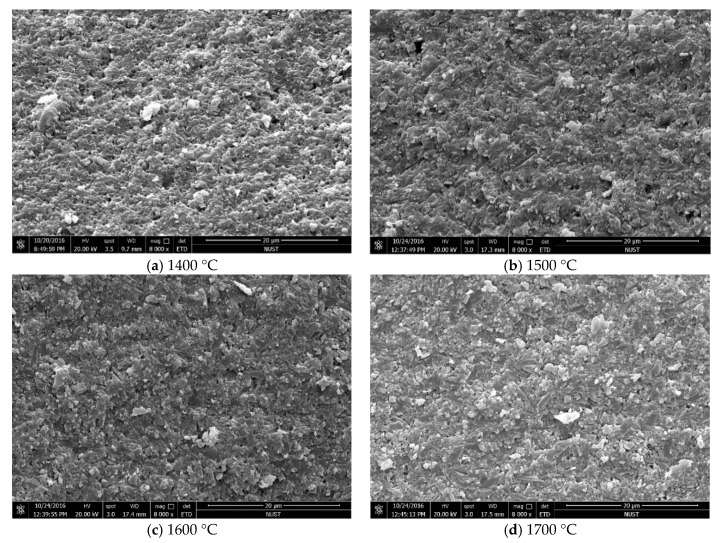
The Scanning electron microscopy (SEM) micrographs of the fracture surface of the sintered Si_3_N_4_/n-SiC ceramics at different sintering temperatures (holding time for 10 min).

**Figure 4 materials-12-03837-f004:**
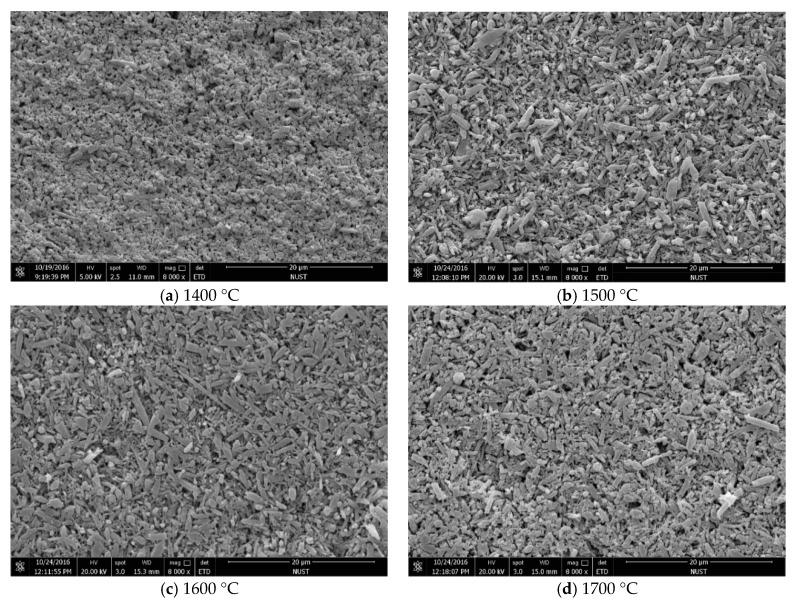
SEM micrographs of the fracture surface morphology of the samples after molten NaOH corrosion at different sintering temperatures (holding time for 10 min).

**Figure 5 materials-12-03837-f005:**
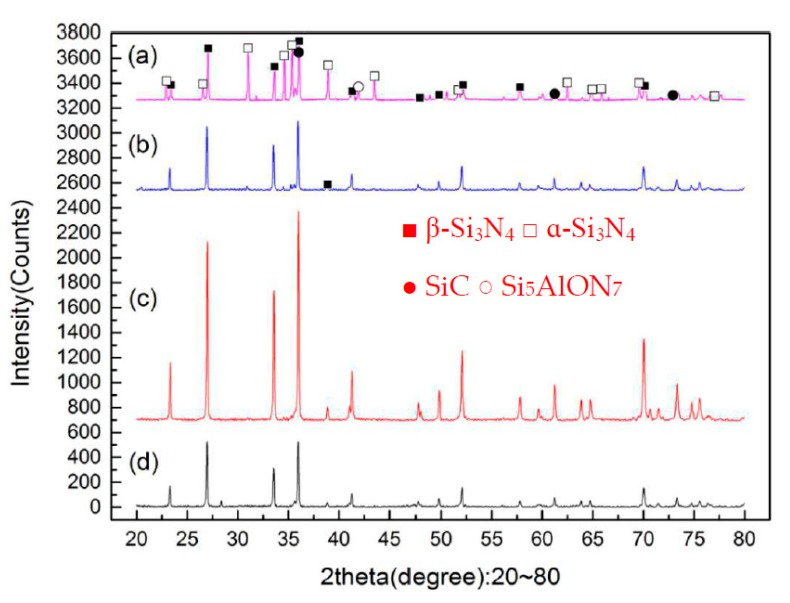
The X-ray diffraction (XRD) results of the sintered Si_3_N_4_/n-SiC ceramics at different sintering temperatures (holding time for 10 min), (**a**) 1400 °C, (**b**) 1500 °C, (**c**) 1600 °C, and (**d**) 1700 °C.

**Figure 6 materials-12-03837-f006:**
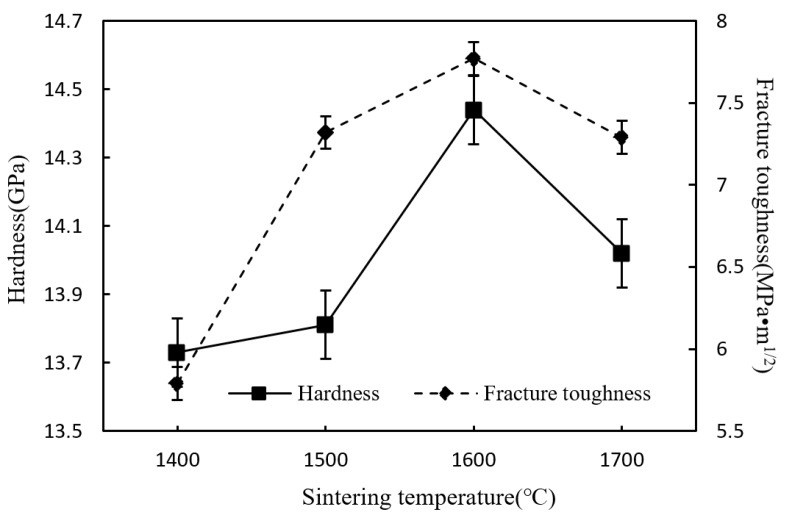
Effect of sintering temperature on hardness and fracture toughness (holding time for 10 min).

**Figure 7 materials-12-03837-f007:**
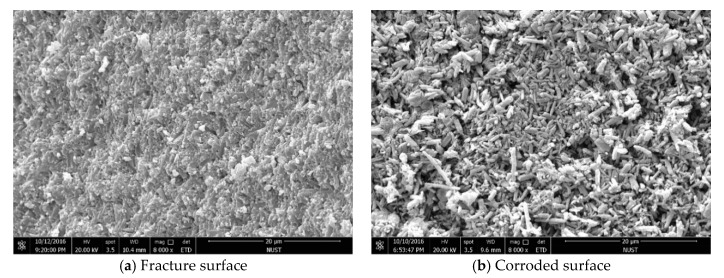
SEM micrographs of fracture surface morphology of the Si_3_N_4_ material (0 wt%) at 1600 °C for 10 min.

**Figure 8 materials-12-03837-f008:**
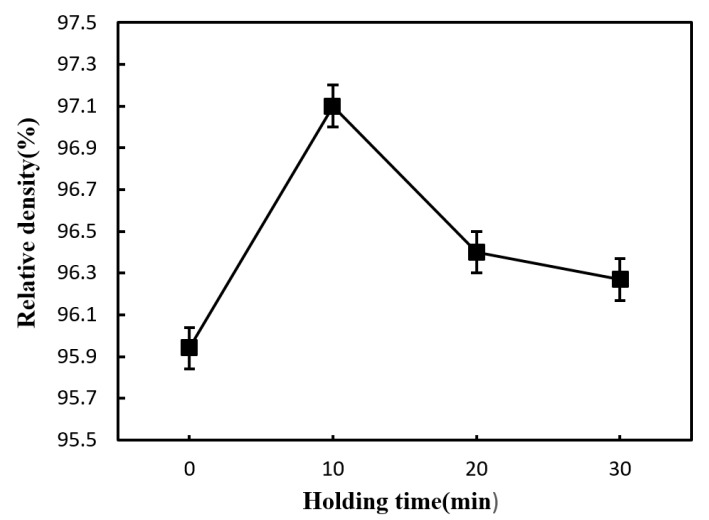
Effect of holding time on the relative density of the sintered Si_3_N_4_/n-SiC ceramic at 1600 °C.

**Figure 9 materials-12-03837-f009:**
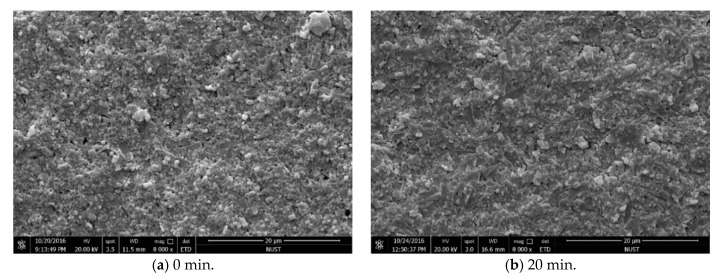
SEM micrographs of fracture surface morphology of the sintered Si_3_N_4_/n-SiC ceramic at different holding time (sintering temperature for 1600 °C).

**Figure 10 materials-12-03837-f010:**
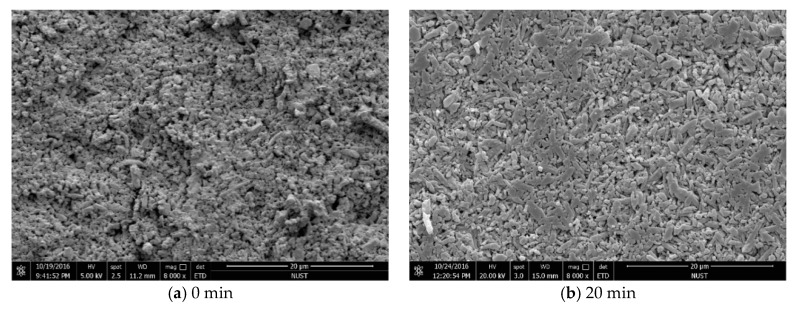
SEM micrographs of the fracture surface morphology of the samples after molten NaOH corrosion at different holding time (sintering temperature for 1600 °C).

**Figure 11 materials-12-03837-f011:**
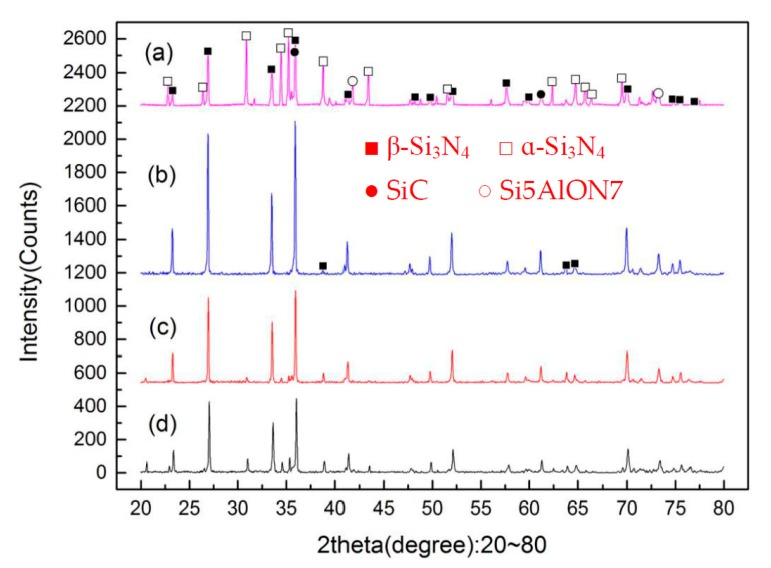
XRD results of the sintered Si_3_N_4_/n-SiC ceramics at different holding time (sintering temperature for 1600 °C), (**a**) 0 min, (**b**) 10 min, (**c**) 20 min, and (**d**) 30 min.

**Figure 12 materials-12-03837-f012:**
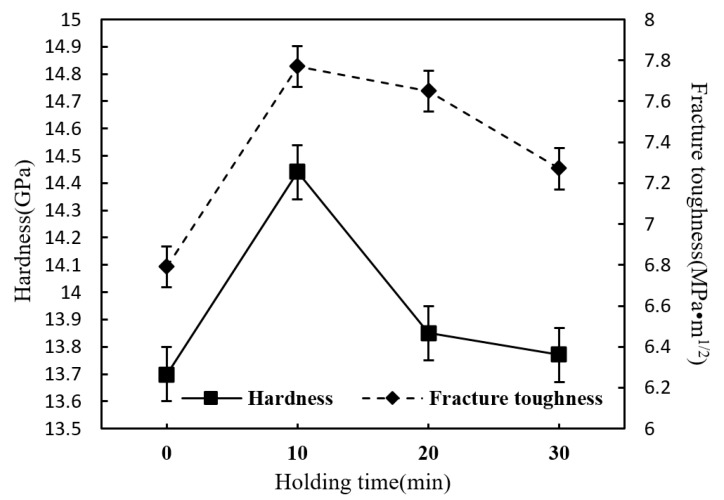
Effect of holding time on hardness and fracture toughness of the sintered Si_3_N_4_/n-SiC ceramics (sintering temperature for 1600 °C).

**Table 1 materials-12-03837-t001:** The sintering temperature and holding time of the series ceramic materials.

Test No.	Compositions/wt%	Temperature/°C	Holding Time/min
1	85% Si_3_N_4_ + 5% SiC + 5% Al_2_O_3_ + 5% Y_2_O_3_	1400, 1500, 1600, 1700	10
2	85% Si_3_N_4_ + 5% SiC + 5% Al_2_O_3_ + 5% Y_2_O_3_	1600	0,10,20,30
3	90% Si_3_N_4_ + 5% Al_2_O_3_ + 5% Y_2_O_3_	1600	10

**Table 2 materials-12-03837-t002:** Comparison of properties of Si_3_N_4_ with SiC nanopowders content of 0% and 5% at 1600 °C for 10 min (wt%).

Number	n-SiC Content/wt%	Relative Densty/%	Hardness/GPa	Fracture Toughness/MPa·m^1/2^
1	0	94.5	13.50	6.87
2	5	97.1	14.44	7.77

**Table 3 materials-12-03837-t003:** Comparison of mechanical properties of silicon nitride.

Additives	Sintering Method	Sintering Temperature/°C	Holding Time/min	Relative Density/%	Hardness/GPa	Fracture Toughness/MPa·m^1/2^
5 wt% Y_2_O_3_ + 5 wt% MgO + 2 wt% Al_2_O_3_ [3]	Microwave	1700	10	98.52	14.92	6.44
3 wt% MgO + 1.5 wt% Al_2_O_3_ + 3.5 wt% SiO_2_ [29]	Pressureless	1780	180	99.70	14.20	6.40
5 wt% MgSiN_2_ + 3 wt% Y_2_O_3_ + 1 wt% CeO_2_ [13]	Spark plasma	1650	6	99.40	16.53	6.89
10 wt% LiYO_2_ + 5 wt% ZrO_2_ [30]	Microwave	1600	15	93.00	13.00	6.10

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
