# Peer review of "Effects of Microwave Sintering Temperature and Holding Time on Mechanical Properties and Microstructure of Si3N4/n-SiC ceramics"

_materials, 2019, doi:10.3390/ma12233837_

Round 1
Reviewer 1 Report
Please improve organisation of text. In many many places there are strange mistakes. For instance in raws 42-50 fragments of instructions for authors were left. In table 1, data of sintering temperatures for sample 2 are mixed with the soaking times.
I am not native English, but I see many language mistakes. Definitely language mus be carefuly revised.
Paper itself gives some experimental data what could be interseted for scientists working on similar topics.
I have only one questions which in my opinion needs at least a probe of explanation: Why composites (data presented in Table 2) which is much better densified than "pure" Si3N4 in not significantly harder than Si3N4? It seems to be strange for me.
Reviewer 2 Report
The article “Effects of Microwave Sintering Temperature and Holding Time on Mechanical Properties and Microstructure of Si3N4/n-SiC ceramics” is an interesting study of the sintering of silicon nitride composite and the application of microwave sintering.
However, this article suffers from a very low level of important details on why the materials (sintering aids) and method are chosen. Before acceptance is a journal like MATERIALS the following major and minor points should be addressed:
Major revisions:
P1 l40 > Why these sintering aids are used? What is their role in the sintering process? This should be clearly specified and justified by literature.
P2 l59 > Very important to specify if the MW sintering is in air, Ar, N2; If in air there is potential for important presence of silica. This should be clear, XRD on sample without NaOH corrosion.
P3 l85 > It is important to justify this liquid phase at these low temperatures (alumina alone >2000°C), is it prealloyed powder? in situ reaction? Local temperature?
P4 l108 > Very important here to specify why NaOH corrosion! The microstructure looks not sintered after this step, Is it the removal of oxide phases because sintered in air?
Minor points:
Page 1
L31 indices in Si3N4
L33 “which should be improved to meet the requirements of high-performance cutting tools” precise which are the required values.
L38 which is the transformation temperature?
P2 l53 > are they commercial powders? precise the ref
ALL XRD figures > it would be better to add the legend of the indexes.
Reviewer 3 Report
I believe that in the article, when solving the research problem, the wrong approach was taken to identify the effects of microwave effects on materials. Be sure to conduct experiments with purely thermal sintering of ceramics. When conducting and discussing experiments with microwave radiation, it is necessary to take into account the issues of radiophysics. The article has not done this. It is well known that with increasing temperature the absorption of microwave radiation increases as the conductivity of materials increases. This means that the Heat preservation device will heat up with different efficiency during microwave heating. These issues are not taken into account.
Nanometer SiC is introduced into the studied ceramics as a material with increased absorption of microwave radiation. At the same time, crucible 4 is completely filled with Silicon carbide powder.
In the microwave field, all the energy is absorbed in this powder. Then the sintered sample will be heated in a thermal manner. Based on this, the question of the existence of radiation heating remains open.
Finally, the authors showed negligence in preparing the manuscript. So lines 42-50 clearly do not relate to the content of the article. Line 79 “As shown in Fig. 79 3(a) shows that..” needs to be adjusted.
English needs to be improved.
Round 2
Reviewer 2 Report
The Authors well respond all the comments.
The article look ready to publish for me.
Reviewer 3 Report
No comments